# Antibiotic-induced changes in the human gut microbiota for the most commonly prescribed antibiotics in primary care in the UK: a systematic review

Karen T Elvers ,[1] Victoria J Wilson,[2] Ashley Hammond ,[2] Lorna Duncan,[2] Alyson L Huntley ,[2] Alastair D Hay,[2] Esther T van der Werf [2,3]

[1]Centre for Academic Primare Care & NIHR Health Protection Research Unit in Behavioural Science and Evaluation, Bristol Medical School, University of Bristol, Bristol, UK
[2]Centre of Academic Primary Care, Bristol Medical School, University of Bristol, Bristol, UK
[3]Department of Integrative Medicine, Louis Bolk Institute, Bunnik, The Netherlands

**Correspondence to**
Dr Esther T van der Werf;
e.vanderwerf@louisbolk.nl

## ABSTRACT

**Objective** The gut microbiota influences many aspects of human health. We investigated the magnitude and duration of changes in gut microbiota in response to antibiotics commonly prescribed in UK primary care.

**Methods** We searched MEDLINE, EMBASE and AMED, all years up to May 2020 including all study designs, collecting and analysing data on the effect of antibiotics prescribed for respiratory and urinary tract infections. We followed the Preferred Reporting Items for Systematic Reviews and Meta-Analyses and Cochrane standard methods. Risk of bias was evaluated using the Critical Appraisal Skills Programme. Narrative synthesis was used to report the themes emerging from the data.

**Main outcome measures** Primary outcomes were antibiotic-induced changes in the composition and/or diversity of the gut microbiota. Secondary outcome was the time for the microbiota to return to baseline.

**Results** Thirty-one articles with low or unclear risk of bias showed that antibiotics impact the gut microbiota by causing rapid and diminished levels of bacterial diversity and changes in relative abundances. After cessation of treatment, gut bacteria recover, in most individuals, to their baseline state within a few weeks. Some studies suggested longer term effects from 2 to 6 months. Considerable heterogeneity in methodology makes the studies prone to biases and other confounding factors. Doxycycline was associated with a marked short-term decrease in *Bifidobacterium* diversity. Clarithromycin decreased the populations of Enterobacteria, and the anaerobic bacteria *Bifidobacterium* sp and *Lactobacillus* sp in numbers and diversity for up to 5 weeks. Phenoxymethylpenicillin, nitrofurantoin and amoxicillin had very little effect on the gut microbiome.

**Conclusions** Despite substantial heterogeneity of the studies and small sample sizes, there is evidence that antibiotics commonly used in primary care influence the composition of the gastrointestinal microbiota. Larger population-based studies are needed to fully understand how antibiotics modulate the microbiota, and to determine if these are associated with (longer term) health consequences.

**PROSPERO registration number** CRD42017073750.

### Strengths and limitations of this study

► This review addresses the antibiotic-induced changes in gut microbiota for the most commonly prescribed antibiotics in UK primary care for the infections most frequently seen in general practice: respiratory tract and urinary tract infections.
► The study used a complete and inclusive search strategy complemented by manually scanning reference lists of identified articles for further relevant publications.
► We searched three databases, all years up to May 2020, for any study type, with clearly defined criteria for study inclusion.
► It was not feasible to combine results in a meta-analysis as the majority of the studies were small, poorly randomised and with limited follow-up. There was considerable heterogeneity in methodology, which makes them prone to biases and other confounding factors.
► This limits our understanding of the long-term changes induced by commonly prescribed antibiotics, the ability to modify antibiotic treatment to each situation and to make recommendations to clinicians.

## INTRODUCTION

In the UK, between 2000 and 2014, antibiotic prescriptions increased from 14.3 to 19.7 defined daily doses (DDDs) per 1000 inhabitants per day. The volume of antibiotic prescribing has since decreased, reaching 18.7 DDDs per 1000 inhabitants per day in 2016 (http://www.oecd.org/). Misuse and overuse of antimicrobials, not only in medical sectors, but also in veterinary and agricultural sectors, result in increasing resistance, which globally limits our ability to control infections. Decreasing antibiotic consumption is a key target to dealing with the problem, and the UK has implemented antimicrobial

stewardship interventions in primary care[1] and detailed a 5-year action plan for tackling antimicrobial resistance.[2]

Data analysed from The Health Improvement Network (THIN) database stated that a median prescribing rate among participating practices was 626 antibiotic prescriptions/1000 patients (IQR 543–699), 69% of which could be linked to a body system and/or clinical condition.[3] Of these prescriptions, 46% were linked to respiratory tract infections (RTIs) and 22.7% to infections of the urogenital tract. About half of all antibiotics prescribed were penicillin, of which approximately 55% was amoxicillin; this is followed by 13% macrolide use and 12% tetracycline.[3] General practicioner (GP) consultation rates in England and Wales show that a quarter of the population will visit their GP because of an RTI each year.[4] Antibiotic use to treat such infection varies between individual clinicians and countries,[4–6] but accounts for 60% of all antibiotic prescribing in general practice.[7–9] Similarly, urinary tract infections (UTIs) are commonly seen in primary care.[10] Patients are frequently treated with antibiotics, even though the infection often is not microbiologically confirmed.[11] The overuse of antibiotics for conditions that are not serious or where antibiotic treatment is not appropriate for the illness has led to their reduced effectiveness and emergence of resistant bacteria.

The normal gut microbiota consists of approximately 800–1000 different bacterial species and more than 7000 different strains.[12] 'Normal' is defined as the commensal species predominantly and consistently found in the gut of healthy human. The composition of the oesophageal and gastric microbiomes is distinct from that of the colon. There is a greater diversity of species and number per gram of contents in the colon, reflecting its higher pH.[13 14] The core microbiome in healthy individuals is dominated by phyla Firmicutes and Bacteroidetes (over 90%), followed by Verrucomicrobia and Actinobacteria.[13] Some essential functions of a healthy gut microbiota include vitamin production, nutrient metabolism, immunomodulation and protection against infection by inhibiting colonisation of the gut by pathogens. Studies into gut microbiomes from different countries have identified differences in diversity, which reflect the differences in diet, geography, early-life exposures and genetics.[15] Age, diet, antibiotics, probiotics and prebiotics all contribute to the shaping of the healthy gut microbiota and this continues dynamically throughout life.[13 14 16]

These differences lead to microbiome dysbiosis, with increasing evidence that alterations to the human microbiome can affect how well the immune system functions and its ability to resist infection[17 18]; increase the risk of developing Parkinson's disease, multiple sclerosis and Alzheimer's disease,[19] the risk of depression, anxiety or psychosis[20]; and indirectly affect the health long term and risk of obesity and diabetes.[21] There is increasing evidence that the microbiome is crucial in a cancer patient's risk of infectious complications.[22] Recent studies have shown that antibiotic exposure in childhood is associated with increased risk for several diseases including obesity, types 1 and 2 diabetes, inflammatory bowel diseases, Crohn's disease, coeliac disease, autoimmune diseases such as allergies, juvenile idiopathic arthritis and asthma.[23–26] It appears that these effects are most pronounced if the antibiotics are consumed within the first 2 years of life and the effects may well be cumulative.[17 27 28] This is particularly important in young children, when the adult microbiome has not been fully established.[29 30] Once antibiotic treatment has stopped, the microbiome may present a certain degree of resilience, being capable of returning to a composition similar to the original one, but this is poorly understood and can take months or even years.[17 31]

Differences in perceptions about how risk-free antibiotic treatment is may in part account for the enormous variation in rates of their use from practitioner to practitioner, between localities and across countries, but also within a specific group of practitioners such as GPs. The real cost of the most commonly prescribed antibiotics in primary care must be clearly understood, including the differences between particular antimicrobial agents in their effects on the microbiome. This information will be specifically useful for antimicrobial stewardship in primary care.

## AIM

This study aimed to investigate the antibiotic-induced changes in gut microbiota for the most commonly prescribed antibiotics in primary care in the UK: those prescribed for RTIs and UTIs. Our primary outcome was defining any antibiotic-induced changes in the composition of the gut microbiome, measured by (1) abundance and/or (2) diversity of the microbes, while our secondary outcome measured the time that is needed to restore the microbiome to a pre-antibiotic state.

## METHODS

A systematic literature review of studies that evaluated the association between antibiotics commonly prescribed for RTI and UTIs in primary care and their effect on the gut microbiome was conducted. Preferred Reporting Items for Systematic Reviews and Meta-analyses (PRISMA) guidelines and checklist were used as a framework for this review (see online supplemental table S1 PRISMA checklist).[32] The study protocol was registered with PROSPERO number CRD42017073750 (http://www.crd.york.ac.uk/PROSPERO).

### Search strategy

Electronic databases MEDLINE, EMBASE and AMED were used to identify relevant articles up to May 2020, in English and available as full text. A combination of search terms included the following keywords: (antibacterial agents OR bacterial infections) AND (microbiota OR microbiome). The strategy was designed to identify studies conducted in any country, investigating antibiotic-induced changes in the human gut microbiota; analysed

## Box 1    Search strategy

1. Anti-Bacterial Agents/ae, pd, tu
2. Bacterial Infections/dt, mi
3. Microbiota/
4. (Microbiome or Microbiomes or Gut microbiome or Gut microbiomes or Microbiota Gut or microbiota).mp
5. Humans/
6. 1 or 2
7. 3 or 4
8. 6 and 7
9. 5 and 8

**Table 1**    PICOS criteria for inclusion and exclusion of studies

| Parameter | Inclusion | Exclusion |
|---|---|---|
| Participants | Human adults and children | Animals, in vitro |
| Intervention | Specific antibiotics of interest prescribed in primary care | Other antibiotics |
| Comparator | Negative controls or other antibiotics | None |
| Outcomes | Changes in gut microbiota, total numbers, diversity and composition; time taken to recover | Antimicrobial resistance |
| Study design | All study types | None |

at either the individual or the population level, published in any language (box 1). Reference lists of included studies and relevant reviews were hand-searched to identify additional eligible studies. All selected abstracts and citations were exported from the scientific databases to the reference management software EndNote X9 (Thompson Reuters, New York, New York, USA) and duplicates excluded.

### Study selection and eligibility criteria
A study was eligible for review if it reported quantitative changes to the gut microbiota due to treatment with antibiotics most commonly prescribed for RTI and UTI prescribed in primary care (table 1). RTIs included both upper respiratory tract infections (URTIs), including the common cold, laryngitis, pharyngitis/tonsillitis, acute rhinitis, acute rhinosinusitis and acute otitis media, and lower respiratory tract infections (LRTIs), including acute bronchitis, bronchiolitis, pneumonia and tracheitis. We included uncomplicated and recurrent urinary tract infections (UTI and rUTIs). We excluded any animal studies or hospital-based studies.

The main exposure of interest in this review was the effect commonly prescribed antibiotics have on the gut microbiome based on prescriptions advice included in the National Institute for Health and Care Excellence (NICE) guidelines[1 8 33–35] and on current use as described in nation reports on antibiotic use.[1 34] Of particular interest are those antibiotics that are commonly used in primary care for the treatment of both upper and lower RTIs (amoxicillin, phenoxymethylpenicillin, doxycycline, co-amoxiclav, erythromycin, clarithromycin) and UTIs (trimethoprim, nitrofurantoin, amoxicillin, cephalexin).

### Screening, data extraction and management
Five reviewers (VW, KTE, EvdW, LD and AH) independently screened search results and assessed each potential study according to the inclusion and exclusion criteria. All decisions were recorded on a spreadsheet. Full-text papers for all eligible studies were obtained and three reviewers (VW, KTE, AH) independently screened the selected papers a second time.

The following data were extracted from the included studies into an Excel spreadsheet by one reviewer (KTE) and a percentage verified for accuracy by another (VW): author, journal, year of publication, study design, study country, number of participants, number of participants of interest to this review, age range, percentage of female participants, other inclusion criteria, exclusion criteria, recruitment methods, recruitment time period, sample type, primary outcome, secondary outcome, antibiotic (of interest to this review) used, any other antibiotics used, dose of antibiotics (of antibiotic of interest to this review), time course (of antibiotic of interest to this review), number of timepoints for collecting samples, time points for each sample collection, testing method, infection or condition, overall conclusion. A tabular summary was developed for this review, which included study ID, country, design, population, setting, infection, antibiotic of interest, sample and analysis, primary outcome and main results (table 2).

### Assessment of methodological quality
Selection bias was assessed with the Critical Appraisal Skills Programme (CASP) checklist (www.casp-uk.net). Each component of the studies (ie, the appropriateness of the study design for the research question, the risk of selection bias, exposure measurement and outcome assessment) was graded and an overall grading used to produce quality assessment charts based on a traffic light system of 'low', 'high' and 'unclear' or 'not applicable', as recommended by Cochrane.[36] Three reviewers (VW, KTE, EvdW) independently assessed the risk of bias in the studies with two reviewers (KTE, VW) assessing all studies, and discrepancies were resolved by consensus between the reviewers.

### Data synthesis and analysis
Due to the insufficient number of studies available, we were unable to undertake any sensitivity analysis to determine robustness of the findings and the quality of the studies.[36] The results are discussed descriptively.

**Table 2** Summary of studies included in the review

| Study ID | Country | Design | Population | Setting | Infection | Antibiotic of interest (and dosage) | Sample and analysis | Primary outcome | Results |
|---|---|---|---|---|---|---|---|---|---|
| **Amoxicillin** | | | | | | | | | |
| De La Cochetière et al (2005)[37] | France | Cohort | N=6 adults | Volunteers | Healthy | Amoxicillin (500 mg three times a day for 5 days) | Faecal samples; 16S rRNA PCR and TTGE gel analysis and sequence analysis | To assess the ability of the human faecal microbiota to return to its original dominant species profile after a 5-day course of amoxicillin | Similarity indices of the TTGE profiles on D1 ranged from 93% to 99%; on D3 62%–82% and D4 46%–94%. On D30 and D60, these had raised back to 65%–95% and 66%–98%, respectively. Bands excised from the gel related to *Clostridium nexile* and *Ruminococcus torques* (and β-Proteobacteria). No normalisation of bacterial abundance D55 after antibiotic treatment. |
| Monreal et al (2005)[38] | Brazil | Prospective | N=42 adults; 20 controls | Emergency department; blood donor centres | Bacterial respiratory infections (sinusitis, pneumonia) | Amoxicillin | Faecal samples analysed by culture and CFU on blood agar | Investigated the influence of respiratory tract infections and of amoxicillin therapy on the normal intestinal microbiota of patients | Concentrations of *Bacteroides*, *Bifidobacterium* spp *Lactobacillus* spp were significantly decreased in the antibiotic group. After 30 days, abundance of *Bifidobacterium* and *Lactobacillus* had normalised. |
| Pallav et al (2014)[39] | USA | RCT | N=24; 8 with amoxicillin, 8 controls, 8 other | Clinical and translational science centre | Healthy | Amoxicillin 250 mg three times a day for 7 days | 16S rRNA gene sequencing region ns Roche 454 GS FLX Titanium GreenGenes/RDP/ NCBI Database | Effect of amoxicillin vs controls | The genera that are most abundant are *Faecalibacterium*, *Bacteroides*, *Rosburia*, *Clostridium* and *Ruminococcus*. No change in bacterial abundance in the control group. The most notable change was an increase in *Escherichia/ Shigella* during antibiotic treatment. Also increase in *Bacteroides* but decrease in *Faecalibacterium* spp. Antibiotic-associated changes persisted to the end of the study, 42 days after antibiotic therapy ended. |
| Abeles et al (2016)[40] | USA | Cohort of unrelated cohabiting individuals randomly assigned | N=56 adults | University campus | Self-reported health status | Amoxicillin 500 mg twice daily for 3 and 7 days | 16S rRNA amplified from DNA using QIAGEN stool Mini Kit; sequencing | Effects of commonly prescribed antibiotics on microbiota | Microbiota grew more dissimilar over time but not significant. Most abundant taxa in gut were Bacteroidaceae, Lachnospiraceae and Ruminococcaceae. Lachnospiraceae, Veillonellaceae, Bacteroidales and Porphyromonadaceae were significantly decreased. Fusobacteriaceae increased. Bifidobacteriales and Erysipelotrichaceae initially decreased. There were sustained reductions in microbiome diversity in response to amoxicillin over 6 months. |
| Mangin et al (2010)[41] | Chile | Study trial | N=42 infants; 31 with treatment | Health centre | Acute bronchitis | Amoxicillin 50 mg/kg/day in three daily doses for 7 days | Faecal; real-time PCR for *Bifidobacterium*, TTGE and sequence analysis | To investigate the quantitative and qualitative changes occurring in the faecal bifidobacterial populations in 18-month children after a 1-week amoxicillin treatment | Total bacteria and numbers of Bifidobacteria were not significantly altered by amoxicillin treatment. However, treatment changed species diversity, a complete disappearance of *B. adolescentis*, increase in *B. bifidum*. No effect on *B. longum*, *B. pseudocatenulatum/B. catenulatum*. |

Continued

**Table 2** Continued

| Study ID | Country | Design | Population | Setting | Infection | Antibiotic of interest (and dosage) | Sample and analysis | Primary outcome | Results |
|---|---|---|---|---|---|---|---|---|---|
| Christensson et al (1991)[42] | Sweden | Randomised double-blind parallel multicentre trial | N=84 adults; 44 with treatment (38/44 completed) | Outpatients | Lower respiratory tract infection | Amoxicillin 250 mg twice daily for 7 days | Faecal; cultured aerobically and anaerobically in broth to ID pathogenic microorganism and on selective plates for analysis of microflora changes | To compare cefaclor and amoxicillin as treatment for lower respiratory tract infections and in their ability to influence colonisation resistance | No change in abundance of *Enterococcus* spp, *Staphylococcus* spp or *Streptococcus* spp, while enterobacteria significantly increased. *Pseudomonas* sp and *Candida albicans* were isolated from some samples. Anaerobic cocci numbers were reduced in some patients and increased in others. Lactobacilli, bifidobacteria, bacteroides and eubacteria increased. A couple of patients were colonised with *C. difficile*. |
| Brismar et al (1993)[43] | Sweden | Single blind randomised trial | N=20 adults; 10 with treatment | Not specified | Healthy volunteers | Amoxicillin 500 mg every 8 hours for 7 days | Faecal; selective media, MICs, beta lactamase activity | To study the oral and intestinal microflora before, during and after administration of cefpodoxime proxetil and amoxicillin | Numbers of aerobic intestinal microflora were slightly affected by amoxicillin administration. A minor decrease in numbers of streptococci and staphylococci; an over growth of *Klebsiella* and *Enterobacter* in a couple of samples. No overgrowth of enterococci or yeasts occurred. Eubacteria was reduced by amoxicillin. Increase in amoxicillin-resistant Enterobactericaeae (*E. coli*, *Klebsiella* sp and *Enterobacter* sp) No change in abundance of *Bacteroides* spp. *Bifidobacterium* spp, *Enterococcus* spp, *Lactobacillus* spp or yeast. No new colonisation with *C. difficile*. Normalisation of bacterial abundance 14 days after antibiotic treatment. |
| Edlund et al (1994)[44] | Sweden | Cohort | N=44 adults, 10 with treatment | Not specified | Not specified | Amoxicillin 500 mg t.i.d for 7 days | Faecal; serial dilutions and selective plating aerobic and anaerobic | The investigation was focused on drug concentrations in faeces, beta lactamase production by the intestinal microflora, alterations in the microflora and susceptibility patterns | Administration of amoxicillin affected the aerobic intestinal microflora to a minor extent. There was an overgrowth of *Klebsiella* spp and *Enterobacter* spp in some samples. No change in abundance of anaerobic bacteria, *Bacteroides* spp, *Bifidobacterium* spp, *Enterococcus* spp, *E. coli*, *Clostridium* spp, *Lactobacillus* spp or yeast. No colonisation with *C. difficile*. Intestinal microflora returned to normal 2 weeks after treatment stopped. |

Continued

**Table 2**  Continued

| Study ID | Country | Design | Population | Setting | Infection | Antibiotic of interest (and dosage) | Sample and analysis | Primary outcome | Results |
|---|---|---|---|---|---|---|---|---|---|
| Ladirat *et al* (2014)[45] | The Netherlands | Double-blinded randomised parallel intervention study | N=12 adults (two not analysed due to non-compliance) | Not specified | Healthy volunteers | Amoxicillin 375 mg three times daily for 5 days | Faecal; Intestinal (I)-Chip microarray, total bacteria and *Bifidobacterium* spp using qPCR | Determined the effects of prebiotic intake on the microbiota of healthy adult subjects during and after treatment with amoxicillin | Total bacteria and *Bifidobacterium* were similar in both groups before treatment. Numbers of *Bifidobacterium* decreased over time due to amoxicillin treatment in both groups, but more so in the antibiotic only group (no prebiotic). After antibiotic treatment, numbers in the antibiotic only group were still lower than with prebiotics. Abundance of *Bifidobacterium* spp, *B. longum* and *B. thermophilum* was significantly higher in the prebiotic group. The composition of the microbiota in most subjects returned to normal after 3 weeks. Due to amoxicillin treatment, a decrease in the abundance of *Bifidobacterium* spp, an overgrowth of Enterobacteriaceae. |
| Floor *et al* (1994)[46] | The Netherlands | Randomised double-blind study | N=80 adults | Recruited from 9 general practices | Chronic bronchitis with purulent sputum | Amoxicillin, 500 mg three times a day for 7 days | Faecal; cultured aerobically and anaerobically on selective media, testing for *C. difficile*, enterococci, staphylococci, yeasts and biochemical identification | Effect of both amoxicillin and loracarbef on oropharyngeal and intestinal microflora | No change in abundance of anaerobic bacteria or *E. faecalis*. Total numbers of aerobic Gram-negative bacteria significantly higher on days 8–10 than at baseline. Enterococci cultured in one patient in amoxicillin group. Yeast numbers increased significantly but returned to baseline on days 21–28. Presence of other aerobic bacteria occurred in a slightly higher percentage of patients in the amoxicillin group. 13/15 new acquired bacteria in the amoxicillin group were *Klebsiella*. No new colonisation with *C. difficile*. Normalisation of bacterial and fungal abundance 21 days after treatment. |
| **Amoxicillin and clavulanic acid** | | | | | | | | | |
| Korpela *et al* (2016)[31] | Finland | Part of a cohort from larger probiotic trial | N=236 children; 142 donated faecal samples | Day-care centres | Antibiotic use for respiratory (88%) other mainly urinary (5%) | Penicillins (amoxicillin with or without clavulanic acid and penicillin V), macrolides (azithromycin and clarithromycin) and sulphonamide–trimethoprim | Faecal by 16S rRNA and sequence analysis; operational taxonomic unit (OTU); culture-based antibiotic sensitivity testing | Use of phylogenetics, metagenomics and antibiotic use on microbiota composition and metabolism | Macrolide use reduced abundance of Actinobacteria and increased *Bacteroidetes* and Proteobacteria. Penicillin groups did not have distinctly different phyla. *Firmicutes* was reduced. The effect of macrolide use was long lasting and was associated with increased risk of asthma and antibiotic-associated weight gain. |

Continued

## Table 2 Continued

| Study ID | Country | Design | Population | Setting | Infection | Antibiotic of interest (and dosage) | Sample and analysis | Primary outcome | Results |
|---|---|---|---|---|---|---|---|---|---|
| Mangin et al (2012)[47] | France | Trial | N=18 treated, no controls | Study centre | Healthy volunteers | 875/125 mg oral dose of amoxicillin/clavulanic acid twice a day for 5 days | Faecal samples, real-time PCR TTGE | Amoxicillin–clavulanic acid treatment on total bacteria and on Bifidobacterium species balance in human colonic microbiota | Total bacterial concentrations as well as bifidobacteria concentrations were significantly reduced after antibiotic treatment. The mean similarity percentages of TTGE bacteria and TTGE bifidobacteria profiles were significantly reduced. Occurrence of B. adolescentis, B. bifidum and B. pseudocatenulatum/B. catenulatum species significantly decreased. Occurrence of B. longum remained stable. Moreover, the number of distinct Bifidobacterium species per sample significantly decreased. Bacterial abundance was not normalised 2 months after antibiotics stopped. |
| Forssten et al (2014)[48] | Finland | Randomised double-blind placebo controlled parallel study | N=80 adults; 40:40 probiotic:placebo | Volunteers | Healthy | Augmentin (875 mg amoxicillin and 125 mg clavulanate) for 7 days | Faecal samples; qPCR for specific microbial groups | To investigate the effect of a specific combination of probiotic strains on the incidence of antibiotic-induced microbiota disturbances | Generally, Lactobacillus and Bifidobacterium were increased/restored in the probiotic group and reduced in the placebo group. Antibiotics reduced levels of Clostridium in both groups, but increased Enterobacteriaceae. |
| Young et al (2004)[49] | USA | Case report | N=1 adult male | Not detailed | Acute sinusitis | Amoxicillin–clavulanic acid (875 and 125 mg, respectively, twice daily for 10 days) | Faecal sample 16S rRNA PCR and sequencing and analysis | Molecular phylogenetic survey of the faecal microbiota from a patient who developed antibiotic-associated diarrhoea during the administration of a broad-spectrum antibiotic | D0,sequences clustered within four bacterial groups: Bacteroides spp, Clostridium sp IV, Clostridium sp XIVa and Bifidobacterium. At D4, Bacteroides still a major component, but B. distasonis group not B. fragilis cluster. No Clostridium sp XIVa or Bifidobacterium spp. 34% were now Enterobacteriaceae. D24, partial reversal, B. fragilis predominated and cluster XIVa returned, no Enterobacteriaceae, no Bifidobacterium spp normalisation of abundance 14 days after treatment. |
| Engelbrekston et al (2009)[50] | USA | Probiotic–antibiotic study arm | N=40 adults | Not specified | Healthy volunteers | Agumentin (amoxicillin and clavulanic acid), 875 mg twice daily for 7 days | Faecal; TRF analysis, PCR and enzyme digest, bacterial culturing on selective media for enumeration of different species | Analysis of faecal terminal restriction fragment length polymorphism data for treatment effects of probiotic treatment concurrent with antibiotic therapy | There was large subject-to-subject variability. Subjects fell into two categories: those with stable baseline microbiota and those where it varied significantly. Antibiotics had a significant effect on faecal microbiota across all subjects. Culture data also had large variation in counts. Increasing trends were visible in Bacteroides and enterics at day 21. There was no trend seen for Clostridium, Bifidobacterium and Lactobacillus. The placebo group had a significant change from baseline at day 21 in numbers of bacteria. |

Continued

**Table 2** Continued

| Study ID | Country | Design | Population | Setting | Infection | Antibiotic of interest (and dosage) | Sample and analysis | Primary outcome | Results |
|---|---|---|---|---|---|---|---|---|---|
| Lode et al (2001)[51] | Germany | Volunteers | N=12 adults | Not specified | Healthy | Amoxicillin/clavulanic acid 1000 mg (750:125 mg, respectively) daily | Faecal; culture on selective agar, colonies counted, isolated and identified to genus level | Investigate the ecological effects of linezolid, compared with those of amoxicillin/clavulanic acid, on the intestinal human microflora | Amoxicillin/Clavulanic acid was associated with significant increase in numbers of enterococci and *E. coli*. Other aerobic bacteria were largely unaffected. Anaerobic *Bifidobacterium* spp, *Clostridium* spp, *Lactobacillus* spp decreased significantly. No change in abundance of *Bacteroides* spp, Gram-positive bacilli or *Klebsiella* spp. *C. difficile* were isolated from 3 of the volunteers. Some resistant enterobacteria such as *E. coli*, *Klebsiella* sp and *Enterobacter* sp were isolated. The microflora was normalised 35 days after the end of administration. |
| Kabbani et al (2017)[52] | USA | Single-centre, open-label, randomised controlled | N=49, 12 antibiotic treated, 12 controls | Clinical and translational science centre | Healthy volunteers | Amoxicillin/clavulanate 875/125 mg twice daily for 7 days | 16S rRNA gene sequencing | To compare and contrast the effects of a probiotic and antibiotic, the main endpoint was change from baseline in the composition of the gut microbiota | Lower diversity (OTUs, Chao index) d10 and d21. Increased abundance of *Parabacteroides* spp persisted 14 days after antibiotic treatment was stopped. Control subjects had a stable microbiota throughout the study period. Significant microbiota changes in antibiotic treatment group included reduced prevalence of the genus *Roseburia* and increases in *Escherichia*, *Parabacteroides* and *Enterobacter*. Microbiota alterations reverted toward baseline, but were not yet completely restored 2 weeks after treatment stopped. |

**Nitrofurantoin**

| Study ID | Country | Design | Population | Setting | Infection | Antibiotic of interest (and dosage) | Sample and analysis | Primary outcome | Results |
|---|---|---|---|---|---|---|---|---|---|
| Stewardson et al (2015)[53] | Switzerland | Prospective cohort | N=40 adults; 10 nitrofurantoin | Ambulatory care | Lower UTI | Nitrofuratonin (100 mg twice daily for 5 days) | Faecal by 16S rRNA and sequence analysis; OTU | Compare the effects of ciprofloxacin and nitrofurantoin on the gut microbiota composition of non-hospitalised patients with UTIs compared with patients without antibiotic exposure and household contacts of patients receiving ciprofloxacin | Ciprofloxacin caused changes in a number of genus of gut bacteria. Substantial recovery after 4 weeks. Nitrofuratonin treatment correlated with a non-significant increase in *Clostridium* spp and decrease in *Faecalibacterium* spp. No change in abundance of *Alistipes* spp, *Bacteroides* spp, *Bifidobacterium* spp, *Blautia* spp, *Dialister* spp, *Eubacterium* spp. *Oscillospira* spp, *Roseburia* spp or *Ruminococcus* spp. |
| Vervoort et al (2015)[54] | Belgium and Poland | Prospective cohort | N=13; 5 controls | Ambulatory patients visiting GPs | Uncomplicated UTI | Nitrofurantoin (100 mg three times daily for 3–15 days) | Faecal; 16S rDNA PCR, sequencing and analysis | The impact of nitrofurantoin treatment on the gastrointestinal flora of patients with uncomplicated UTIs | Nitrofurantoin treatment did not significantly impact on the faecal microbiota other than a temporary increase in the Actinobacteria phylum (*Bifidobacterium* spp). No change in abundance of Bacteroidetes, Firmicutes, Proteobacteria, Tenericutes or Verrucomicrobia. Normalisation in abundance of Actinobacteria d31–43 after treatment. |

Continued

**Table 2** Continued

| Study ID | Country | Design | Population | Setting | Infection | Antibiotic of interest (and dosage) | Sample and analysis | Primary outcome | Results |
|---|---|---|---|---|---|---|---|---|---|
| Mavromanolakis et al (1997)[55] | Greece | Randomised | N=21 women; 7 with treatment | Not specified | UTI (at least three episodes caused by Enterobacteriaceae in the preceding 12 months) | Nitrofurantoin (100 mg daily for 30 days) | Faecal samples (and urine) cultured and plated on selective agar, API test kits for identification | Impact of doses of norfloxacin, trimethoprim-sulfamethoxazole and nitrofurantoin on aerobic bowel flora | Before antibiotic treatment, all stools contained Enterobacteriaceae and Enterococcus spp. Nitrofurantoin did not alter abundance of Enterobacteriaceae or Enterococcus spp in the colonic flora during treatment. No increased resistance to nitrofurantoin in Gram-negative aerobic bacteria. No patient receiving the drug was found to be colonised by yeasts. |
| **Doxycycline** | | | | | | | | | |
| Heimdahl & Nord (1983)[56] | Sweden | Trial | N=10 | Not specified | Healthy volunteers | Doxycycline 100 mg daily for 7 days | Faecal culture on selective media | Effect of doxycycline on the normal human flora and on colonisation of the oral cavity and colon | No change in abundance of Bacteroides spp, Bifidobacterium spp, Clostridium spp, Eubacterium spp. Lactobacillus spp or Veillonella spp. Fusobacteria were eliminated. A 2–3 log decrease in enterococci, streptococci and enterobacteria. Emergence of new strains Klebsiella pneumoniae, Proteus mirabilis and Enterobacter cloacae in some subjects. All colonising microorganisms were resistant to doxycycline. Normalisation of aerobic bacterial abundance 9 days after antibiotic treatment. |
| Walker et al (2005)[57] | USA | RCT | N=69 adult; 55 analysed | Clinic | Periodontitis | 20 mg doxycycline | Faecal samples plated on a number of selective agar and incubated for CFU counts | To determine if a 9-month regimen of suboptimal doxycycline had an effect on either the intestinal or the vaginal microflora | The only statistically significant differences between the two treatment groups occurred in the doxycycline-resistant counts at the baseline sample. No between-treatment differences were detected at 3-month or 9-month period either in the predominant bacterial taxa or in antibiotic susceptibilities |
| Matto et al (2008)[58] | Finland | Not specified | N=19; 10 controls | Not specified | Not specified | Doxycycline 150 mg daily for 10 days with probiotic | Faecal 16S rRNA PCR for Bifidobacterium sp and DGGE and culturing and sequencing | To evaluate the influence of doxycycline therapy on the composition and antibiotic susceptibility of intestinal bifidobacteria | Bifidobacterium diversity was markedly higher in the control group than antibiotic group; each subject had 2–3 genotypes in the control group; 0–3 in the antibiotic group. The isolated bifidobacteria represented B. adolescentis/ B. ruminantium, B. longum, B. catenulatum/ B. pseudocatenulatum, B. bifidum and B. dentium. Tetracycline-resistant Bifidobacterium isolates were more commonly detected in the antibiotic group than in the control group, thus increasing the pool of resistant commensal bacteria in the intestine. |

Continued

**Table 2** Continued

| Study ID | Country | Design | Population | Setting | Infection | Antibiotic of interest (and dosage) | Sample and analysis | Primary outcome | Results |
|---|---|---|---|---|---|---|---|---|---|
| Rashid et al (2013)[59] | Sweden | Double-blind, randomised, placebo-controlled, parallel group study | N=34, 17 treated, 17 controls | Clinical trial unit | Healthy volunteers | Doxycycline 40 mg capsules orally once daily | Culture on selective media aerobic and anaerobic | Primary objective of this study was to assess the impact of antimicrobial treatment on the oropharyngeal and intestinal microflora during and after administration of 40 mg doxycycline capsule given once daily to healthy volunteers | Doxycycline was detectable in stool up to 16 weeks. No changes in abundance (>2 log CFU/g) of *Bacteroides* spp, *Bifidobacterium* spp, *Clostridium* spp, *Candida* spp, *Lactobacillus* spp or Enterobacteriaceae anaerobic intestinal microflora. No new colonisation with *C. difficile*. At week 20, the anaerobic microflora was normal. In aerobic intestinal microflora, there were changes (2 log CFU/g) in the numbers of enterococci and *E. coli* during the 16-week treatment. Other microorganisms such as other enterobacteria, *Candida* spp and other microorganisms were not affected. The aerobic microflora was normal at week 20. Increase in doxycycline resistance in *Bifidobacterium* spp, anaerobic cocci and Gram-positive rods. |
| **Clarithromycin** | | | | | | | | | |
| Brismar et al (1991)[60] | Sweden | Cohort study | N=10 adults | Not specified | Healthy volunteers | Clarithromycin 250 mg twice daily for 7 days | Culture on selective and non-selective agar | To compare the effect of clarithromycin and erythromycin on the normal flora | Clarithromycin decreased the numbers of *Bacteroides* spp, *Bifidobacterium* spp, *Lactobacillus* spp and *Streptococcus* spp. Anaerobic cocci increased and *C. albicans* was isolated from two patients. No changes in the abundance of *Bacillus* spp, *Clostridium* spp, *Corynebacterium* spp, *Enterococcus* spp, *Eubacterium* spp, *Micrococcus* spp, *Staphylococcus* spp or *Veillonella* spp. Normalisation in abundance of Enterobacteriaceae, *Lactobacillus* spp and *Streptococcus* spp but not of *Bacteroides* spp and *Bifidobacterium* spp 14 days after antibiotic treatment. |

Continued

**Table 2** Continued

| Study ID | Country | Design | Population | Setting | Infection | Antibiotic of interest (and dosage) | Sample and analysis | Primary outcome | Results |
|---|---|---|---|---|---|---|---|---|---|
| Edlund *et al* (2000)[61] | Sweden | Healthy volunteers | N=12 male adult | Not specified | Healthy | Clarithromycin 500 mg twice daily for 7 days | Faecal; inhibition assay, culture on selective and non-selective agar | To investigate the ecological effects of moxifloxacin with those of clarithromycin on the intestinal human microflora | Clarithromycin significantly reduced *E. coli* in numbers during treatment but returned to pretreatment levels at day 35. Six subjects were colonised by resistant aerobic Gram-negative rods, *Citrobacter, Klebsiella, Proteus* and *Pseudomonas*. No overgrowth of yeasts. Total number of anaerobic bacteria decreased during treatment. Bifidobacteria were eliminated or strongly reduced. *Bacteroides* declined. Lactobacilli and clostridia were markedly declined but not significantly. Only minor alterations in numbers of peptostreptococci, veillonellacoci. No overgrowth of *C. difficile*. Bacterial abundance was normalised 28 days after antibiotic treatment. |
| Edlund *et al* (2000)[62] | Sweden | Randomised double-blind controlled study | N=20 adults | Not specified | Healthy | Clarithromycin 500 mg for 10 days | Faecal; culture on selective agar aerobic and anaerobic, Gram stain and biochemical tests, gas liquid chromatography | Assess the impact of antibiotic on intestinal microflora | Moderate disturbances in aerobic intestinal microflora, number of *E. coli* significantly reduced at day 10. Overgrowth of *Klebsiella, Citrobacter* and *Enterobacter* spp occurred in six subjects receiving clarithromycin. No significant overgrowth of *Candida* spp. There was a marked reduction in *Bifidobacterium* spp, *Clostridia* spp, *Lactobacillus* spp, *Micrococcus* spp and *Staphylococcus* spp. No change in the total numbers of anaerobic bacteria, *Enterococcus* spp, *Peptostreptococcus* spp, *Bacteroides* spp, *Candida* spp, *Streptococcus* spp or *Veillonella* spp. The lower numbers of *Bifidobacterium* spp and *Lactobacillus* spp persisted even after 14 days of no antibiotics. |
| Matute *et al* (2002)[63] | The Netherlands | Double blind-randomised trial | N=18, 6 treatment; 6 controls | Medical centre | Healthy volunteers | Clarithromycin 500 mg twice a day orally for 7 days | Faecal culture on selective agar aerobic and anaerobic | To compare the effect of 3-day, twice daily clarithromycin and with placebo on the faecal microflora | No change in abundance of *Candida* spp or *Enterococcus* spp. No new colonisation with *C. difficile* or non-fermenters. No increase in clarithromycin-resistant bacteria. The number of organisms of the family Enterobacteriaceae decreased slightly after antibiotic, but levels normalised by day 21 after therapy. The total number of anaerobic bacteria was not affected in the study group. |

**Phenoxymethylpenicillin**

Continued

**Table 2** Continued

| Study ID | Country | Design | Population | Infection | Setting | Antibiotic of interest (and dosage) | Sample and analysis | Primary outcome | Results |
|---|---|---|---|---|---|---|---|---|---|
| Heimdahl & Nord (1979)[64] | Sweden | Trial | N=20, 10 with phenoxymethylpenicillin | Healthy volunteers | Research Institute | Phenoxymethylpenicillin 800 mg as a loading dose, 800 mg twice daily for 7 days | Faecal culture on selective media under aerobic and anaerobic conditions. Bacterial identification by API strips | Effect of phenoxymethylpenicillin (and clindamycin) on the aerobic and anaerobic microflora in the human mouth, throat and colon | No change in abundance of *Acidaminococcus* spp, *Bacteroides* spp, *Bifidobacterium* spp, *Clostridium* spp, Enterobacteriaeae, *Enterococcus* spp, *Eubacterium* spp, *Fusobacterium* spp, *Lactobacillus* spp, *Megashaera* spp, *Streptococcus* spp or *Veillonella* spp |
| Adamsson et al (1997)[65] | Sweden | Healthy volunteers | N=20 adults; 10 with treatment | Healthy | Not specified | Phenoxymethylpenicillin 1000 mg twice a day for 10 days | Faecal antimicrobial assay and culture | To investigate the ecological effects of phenoxymethylpenicillin, on the oropharyngeal and intestinal human microflora | No change in abundance of *Bacillus* spp, *Bacteroides* spp, *Enterococcus* spp or *Streptococcus* spp. There were minor alterations in numbers of *E. coli*. Three subjects became colonised with *Klebsiella* and one with high numbers of non-fermentative gram-negative rod. Numbers of *Clostridium* increased. No change in total numbers of aerobic or anaerobic bacteria. Normalisation of bacterial abundance day 14 after antibiotic treatment. |
| **Erythromycin** | | | | | | | | | |
| Heimdahl & Nord (1982)[66] | Sweden | Trial | N=10 | Healthy volunteers | Research institute | Erythromycin stearate was given orally in doses of 500 mg twice daily for 7 days | Faecal specimens were taken up to 16 days for cultivation of aerobic and anaerobic bacteria | The impact of erythromycin administration on the normal human flora and on colonisation of the oral cavity, throat and colon | Suppression of both aerobic and anaerobic faecal flora occurred. All subjects were colonised by erythromycin-resistant enterobacteria, clostridia or yeasts in the colon. In aerobes, the number of enterobacteria, enterococci and streptococci were reduced. In anaerobes, no change in abundance of *Bifidobacterium* spp, *Eubacterium* spp or *Lactobacillus* spp. However, *Bacteroides*, fusobacteria and *Veillonella* were eliminated in some subjects. |
| Brismar et al (1991)[60] | Sweden | Cohort study | N=10 adults | Healthy | Not specified | Erythromycin 1000 mg twice daily for 7 days | Faecal culture on selective media | Compare the effect of clarithromycin and erythromycin on intestinal microflora | Changes in the intestinal aerobic and anaerobic microflora. Streptococci eliminated, enterobacteria strongly suppressed. Minor reduction in enterococci and corynebacterial. Staphylococci and *C. albicans* increased. Normalisation in abundance of *Bifidobacterium* spp and *Lactobacillus* spp 14 days after treatment. Anaerobic, lactobacilli, bifidobacteri, clostridia and bacteroides reduced. No normalisation in abundance of *Bacteroides* spp and *Clostridium* spp 14 days after antibiotic treatment. |

CFU, colony-forming unit; GPs, general practitioners; RCT, randomised controlled trial; TTGE, temporal temperature gradient gel electrophoresis; UTI, urinary tract infection.

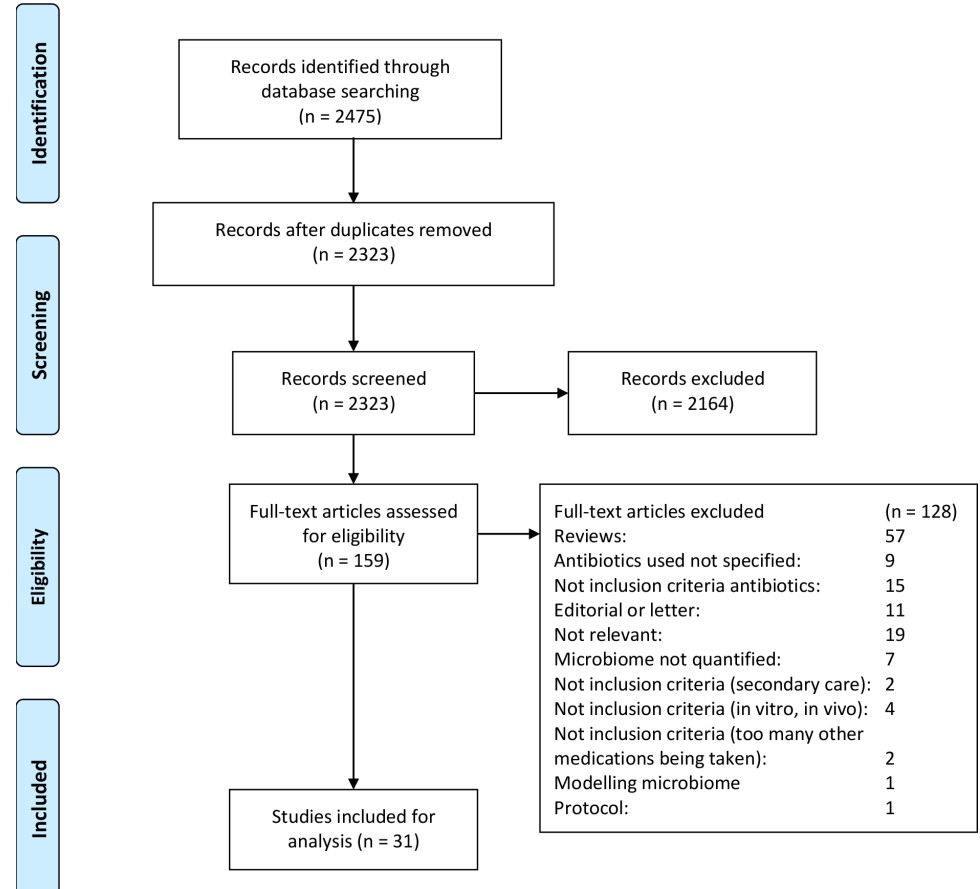

**Figure 1** Preferred Reporting Items for Systematic Reviews and Meta-Analyses flowchart for study selection.

### Patient and public involvement
This research was done without patient involvement.

## RESULTS
### Overall
The database searches returned 2475 articles (May 2020). Following removal of duplicates and screening of records by title or abstract, 2164 were excluded, based on our selection criteria (figure 1). One hundred and fifty-nine full-text articles were evaluated and a further 128 were excluded with reasons detailed in figure 1. Thirty-one studies were eligible and included in the present review. The included studies are summarised in table 2 and are briefly described below.[31 37–66] Twelve of the studies were published by different authors from the same Swedish study group, although one of the studies was carried out in Germany.[42–44 51 56 59–62 64–66]

Publication dates ranged from 1979 to 2017. The studies measured the effects on gut microbiota following clinical treatment of the patient or experimental exposure of volunteers to different antimicrobial agents.

The 31 publications involved 1068 patients, although not all of these received the antibiotics of interest and some were comparative control groups. Most studies originated from European countries, Sweden (11), Finland (3), The Netherlands (3), France (2) and one each for, Germany, Switzerland, Belgium/Poland and Greece. The remaining studies were conducted in the USA (6), Chile (1) and Brazil (1).

## STUDY CHARACTERISTICS
### Sampling
Faecal samples have generally been accepted as representative of the intestinal microbiota. Studies conducted before 2000 used cultured homogenised faecal samples on non-selective and selective media to enumerate different colony types. Some isolated pure cultures for identification to genus level. The effect of the antibiotic was measured by comparing the numbers of bacteria present from each genus before treatment with the decrease, increase overgrowth or proliferation of other bacteria. However, it is known that many types of gut bacteria, about 80%, have not been cultured.[67] Predominantly, later studies used more rapid molecular methods.

### Analytical techniques
All studies analysed faecal samples, either by culture (17 studies) or molecular techniques (13 studies), one study used both methods. Eighteen of the studies used healthy volunteers, four of the studies looked at patients with a respiratory infection, three studies patients had a UTI, one study reported patients with urinary and respiratory

infections, one study patients had sinusitis, one study they had periodontitis, another study relied on self-reported infections and two studies did not specify the health status of the subjects.

### Risk of bias
The risk of bias among the studies was assessed using the CASP checklists for case controls, randomised controlled trials (RCTs), other trials and cohort studies (figure 2). There was 1 case control appraisal, 15 cohort studies and 15 trials. All the studies adequately stated their study aims and hypothesis and this part of the assessment was rated as 'low' risk.

### Case control
There was one case control study included in our review of a patient presenting with onset of diarrhoea not associated with *Clostridium* infection, within 24 hours of antibiotic administration.[49] It was shown that this was associated with changes in the diversity of the gut microbiota. In terms of bias, the study trends toward overall unclear risk as there is lack of controls and the report details just one patient in a clinical setting with a symptom of interest (figure 2A). However, they did use very robust molecular methods to study these temporal changes in the diversity of the microbiota.

### Cohorts
There were 15 cohort studies included in our review. Assessment relating to the validity of the study is covered within questions 1–6 (figure 2B). Generally, these studies were of low risk. The reliability of the studies determined by the size of the effect and how precisely this was estimated are covered with questions 7–9 and are of low or unclear risk. There is greater unclear risk when it comes to the applicability of the results (questions 10–12), the usefulness for clinical decision-making, often the population size was too small to extrapolate results to a larger population as a whole.

### Trials
Fifteen studies included in this review were trials/RCTs and overall the risk was more unclear. The validity of the studies (Q1-6), randomisation and masking of treatment allocation was generally of low risk. The baseline characteristics of the study were of low or unclear risk as most studies stated age, gender and health status, a few exceptions stated body mass index and ethnicity. Follow-up of study participants was of low risk, with most studies taking samples up to 4 weeks after antibiotic administration (although there are some indications that this needs to be longer). Controlling for confounding factors (Q6) was unclear in most of the studies. The size of treatment effect (Q7) was well reported, but precision (Q8) was unclear, mostly due to small population size and lack of statistical analysis. The transferability of the results to other populations (Q9 and 11) was of unclear risk.

### Antibiotic effects on the gut
The studies analysed employed different methodologies to explore the effect of antibiotics on the gut microbiota. They have been consistent in demonstrating that dysbiosis develops on antibiotic administration, which is rapid and results in losses of diversity and shifts in abundance of gut microbes either up or down. After antibiotic treatment, the composition generally returns to a similar pretreatment state within several weeks, but not in all cases. Here, we describe the results in more detail per antibiotic of interest.

### Amoxicillin
Ten studies examined the impact of orally administered amoxicillin on the normal intestinal microflora and are summarised in table 3.[37–46] In brief, administration of amoxicillin leads to an increase or overgrowth of Enterobacteria (six studies). The changes in the anaerobic population (includes *Lactobacillus, Bifidobacterium* and *Bacteroides*) varied enormously between studies: from no change, to increases and/or decreases, as well as changes in diversity. There were two reports that isolated *Clostridium* and *Candida* and two where no change in numbers were observed. Eight of the studies had some longer term follow-up and most of these reported that the populations returned to normal within 2–4 weeks. Three of the studies reported that significant alterations in the microbiome diversity or abundance were still evident from 42 days to 6 months after the final antibiotic dose. Other longer term effects were a persistence of *Candida* at 6 weeks, appearance of other aerobic or resistant Gram-negative bacteria (species not specified) at 28 days, and a persistent decrease in Lachnospiraceae at 6 months, with no similarities in the populations treated.

Christensson *et al*[42] conducted a culture-dependent randomised study of 84 subjects with lower respiratory tract infections, 38 of whom completed treatment with amoxicillin (the remaining received cefaclor and will not be discussed further here). They showed that the numbers of enterobacteria, anaerobic Gram-positive Lactobacilli, Bifidobacteria and Eubacteria, and anaerobic Bacteroides increased significantly in the amoxicillin group. The number of anaerobic cocci was reduced in 18 patients and increased in 11 patients. There was no change in the abundance of *Enterococcus* spp, *Staphylococcus* spp or *Streptococcus* spp. Four patients became colonised with *Pseudomonas* sp and five with *C. albicans*. Although *C. albicans* was also isolated from two patients pretreatment and three patients 6 weeks after treatment *C. difficile* was isolated from two patients during treatment.

The effect of amoxicillin on the intestinal microflora of patients with bronchitis was evaluated in a randomised study by Floor *et al*.[46] The anaerobic flora remained unchanged, both during and after therapy with amoxicillin. The total number of aerobic Gram-negative rods increased significantly, and in 37.5% of the patients with newly acquired bacteria, the main species identified were *Klebsiella*. Enterococci were cultured in only one patient.

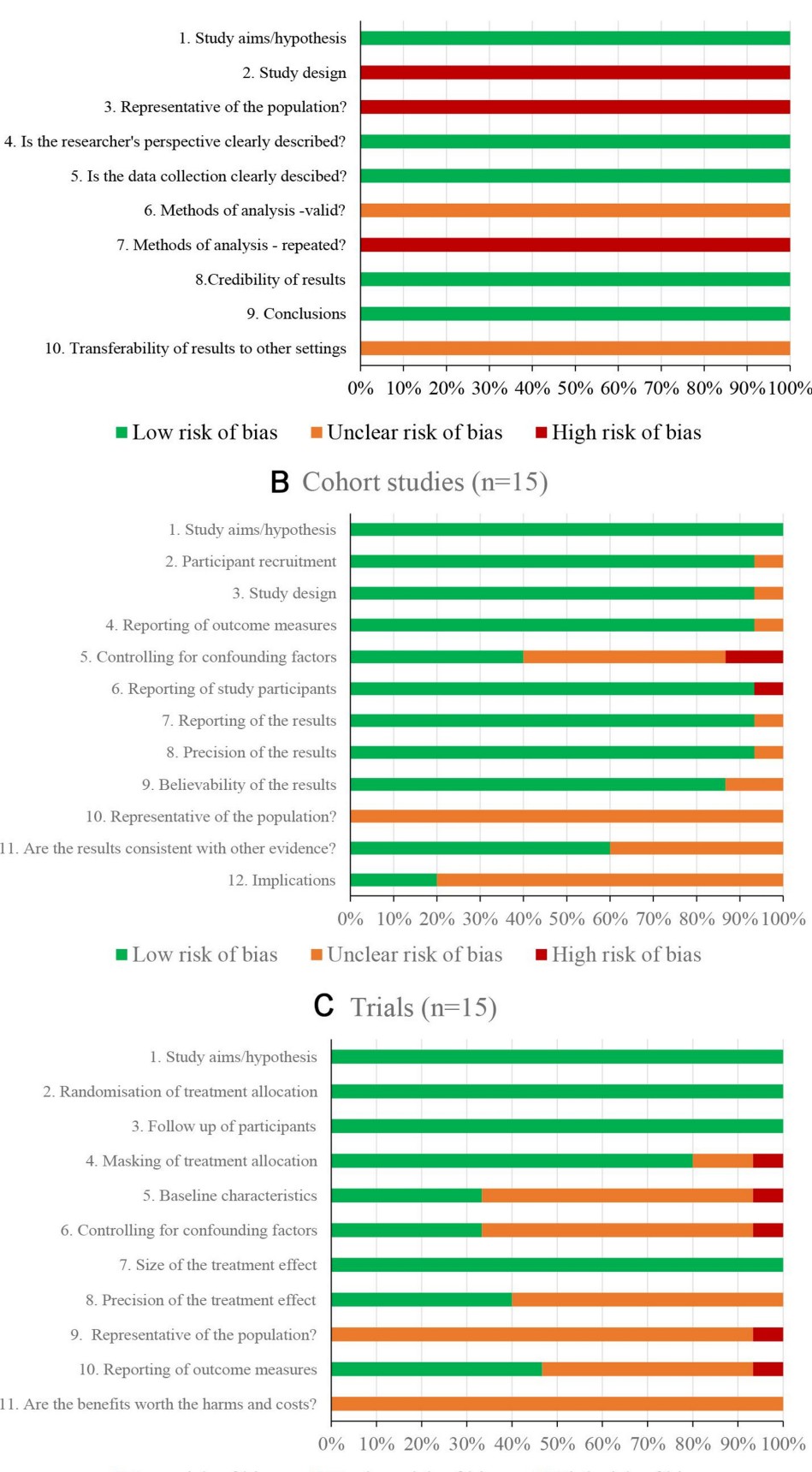

**Figure 2** Overall risk of bias assessment, for case studies (A), cohort studies (B) and trials (C).

**Table 3** Impact of amoxicillin on the gastrointestinal microflora

| Study | Dose (mg/day) | Days of administration | Number of patients | Impact on | | | | | | | Overgrowth | |
|---|---|---|---|---|---|---|---|---|---|---|---|---|
| | | | | Aerobic GP cocci | Pseudomonas | Enterobacteria | Anaerobic | Lactobacillus | Bifidobacterium | Bacteroides | Clostridium | Candida |
| Christensson et al (1991)[42] | 250×3 | 7 | 38 | No change | isolated 4 patients | Increase | – | Increase | Increase | Increase | Isolated two patients | Isolated 10 patients |
| Floor et al (1994)[46] | 500×3 | 7 | 40 | Slight change | – | Increase | No change | – | – | – | – | Increase |
| Mangin et al (2010)[41] | 50/kg/day | 7 | 31 | – | – | – | – | – | Changes in diversity | – | – | – |
| Monreal et al (2005)[38] | – | – | 22 | – | – | – | – | Decrease | Decrease | Decrease | – | – |
| Ladirat et al (2014)[45] | 375×3 | 5 | 12 | – | – | Overgrowth | – | – | Decrease | – | – | – |
| Pallav et al (2014)[39] | 250×3 | 7 | 8 | – | – | Increase | Decrease | – | – | Increase | – | – |
| Brismar et al (1993)[43] | 500×3 | 7 | 10 | Minor decrease | – | Increase | No change | No change | No change | No change | None | No change |
| Edlund et al (1994)[44] | 500×3 | 7 | 10 | – | – | Overgrowth | – | No change | No change | No change | No change | No change |
| De La Cochetiere et al (2005)*[37] | 500×3 | 6 | 5 | – | – | – | – | – | – | – | – | – |
| | | | | | | | Fusobacteriaceae | Lachnospiraceae | Bifidobacteria/ Erysipelotrichaceae | Bacteroidales/ Veillonellaceae/ Porphyromonadaceae | | |
| Abeles et al (2016)†[40] | 500×2 | 3/6 | 6/6 | – | – | – | Increase | Diminished even after 6 months | Decrease then increase | Decrease | – | – |

*This paper measured similarity indices of TGGE profiles and showed shifts in dominant species on antibiotic treatment.
†Also a reduction in diversity.
-, not reported; GP, gram positive; TTGE, temporal temperature gradient gel electrophoresis.

A third of patients had other aerobic bacteria different from baseline on days 21–28. Bacterial and fungal abundance returned to baseline levels 21 days after treatment.

Mangin *et al*[41] studied 18-month old children with acute bronchitis. Both total bacteria and numbers of Bifidobacteria were similar during the course of antibiotic treatment. However, changes in *Bifidobacterium* at the species level showed decreased diversity. Amoxicillin treatment induced a complete disappearance of *Bifidobacterium adolescentis* species, a significant decrease in the occurrence rate of *Bifidobacterium bifidum*, but did not affect *Bifidobacterium longum* and *Bifidobacterium pseudocatenulatum/Bifidobacterium catenulatum*. They did not study longer term effects.

Monreal *et al*[38] used culture-based methods to examine the populations of faecal bacteria in 22 patients with RTIs treated with amoxicillin. *Bacteroides* spp, *Bifidobacterium* spp and *Lactobacillus* spp were decreased with antibiotic treatment. However, this was only transient, and 30 days after the end of treatment, the levels of *Bifidobacterium* and *Lactobacillu*s had recovered to their normal values. *Bacteroides* were slower to recover.

Ladirat *et al*[45] studied a small number of healthy volunteers with amoxicillin and a prebiotic specifically looking at total bacteria and *Bifidobacterium*. Numbers of *Bifidobacterium* decreased over time due to amoxicillin treatment, especially in the group with no prebiotic. Amoxicillin affected the abundance and diversity of *Bifidiobacterium* spp and there was an overgrowth of Enterobacteriaceae. The observations were highly individual dependent. In the follow-up period (3 weeks after the discontinuation of amoxicillin treatment), the levels of bifidobacterial in faeces returned to the initial levels.

Amoxicillin was administered randomly to a small number of healthy volunteers and the impact on intestinal microflora (and oropharyngeal) was studied by Brismar *et al*.[43] In the aerobic intestinal flora, a minor decrease in the numbers of streptococci and staphylococci was observed. Overgrowth of *Klebsiella* species was seen in six subjects and of *Enterobacter* species in two subjects. No overgrowth of enterococci or yeasts occurred. In the anaerobic microflora, only the number of eubacteria was reduced. There was no colonisation by *C. difficile*. Two weeks after administration of amoxicillin, the microbiota had returned to baseline levels.

Another study on the effect of amoxicillin on intestinal microflora of healthy volunteers showed only small alterations in the aerobic faecal microflora.[44] All major anaerobic bacterial groups were unaffected by amoxicillin and there was no colonisation by *C. difficile*. There was an overgrowth of *Klebsiella* spp in six subjects and *Enterobacter* in two subjects. Intestinal microflora returned to normal 2 weeks after treatment had been discontinued.

Six healthy volunteers received oral amoxicillin for 5 days and their faecal microbiota examined by 16S rRNA gene amplification and gradient gel electrophoresis until 60 days.[37] Dominant species diversity profiles were compared on the basis of similarity with percentages on first sampling ranging from 93% to 99%. During the 5-day course of amoxicillin treatment, these percentages decreased to an average of 73%, but there was huge variation between individuals. By day 60, five of the six subject gel profiles had returned to their near initial composition. Gel banding patterns were examined in more detail for three individuals and were shown to correspond to *Clostridium nexile*, *Ruminococcus torques* and β-*Proteobacteria*.

Abeles *et al*[40] studied cohabiting individuals over 6 months where one was given an antibiotic and the other a placebo. The microbiota of subjects taking amoxicillin grew more dissimilar over time, although not statistically significant, as similar trends were also seen in those subjects taking placebo. Taxonomic compositions of the gut differed after amoxicillin therapy. Lachnospiraceae were significantly diminished and remained diminished at 6 months. Veillonellaceae, Bacteroidales and Porphyromonadaceae were significantly decreased in response to amoxicillin, while Fusobacteriaceae were increased. *Bifidobacteriales* and Erysipelotrichaceae were initially decreased and subsequently increased in comparison to their housemates taking placebo.

Pallav *et al*[39] studied the effect of amoxicillin treatment with and without a prebiotic. Eight subjects that received just the antibiotic had substantial microbiome changes, most notably an increase in *Escherichia/Shigella*. There was no change in abundance in the control group. Antibiotic-associated changes persisted to the end of the study, 42 days after antibiotic therapy ended.

## Amoxicillin with clavulanic acid

Seven studies have looked at the impact of amoxicillin in combination with clavulanic acid on the gut microbiota.[31 47–52] These are summarised in table 4. Briefly, the combination of amoxicillin with clavulanic acid caused mild to moderate changes in microbiome composition, mainly increases in Enterobacteria with varying effects on anaerobic bacteria *Bifidobacterium* sp, *Lactobacillus* sp and *Bacteroides* sp and a general decrease in diversity. One study reported that bacterial abundance was not normalised 2 months after antibiotic treatment was stopped, and four reported levels similar to baseline within 35 days.

Administration of amoxicillin/clavulanic acid to healthy people caused increased numbers of enterococci and *Escherichia coli* strains in the aerobic microflora, while Bifidobacteria, Lactobacilli and Clostridia decreased significantly.[51] Numbers of anaerobic cocci and *Bacteroides* were not markedly altered. *C. difficile* strains were recovered from three of the volunteers. The microflora was normalised in all volunteers after 35 days.

Young and Schmidt[49] investigated the short-term impact of amoxicillin/clavulanic acid prescribed for a male patient with acute sinusitis, who developed antibiotic-associated diarrhoea. It was shown that the major bacterial groups were partially restored 14 days after antibiotic treatment, except for *Bifidobacterium*. During antibiotic administration, no sequences corresponding to butyrate-producing *Clostridium* cluster XIVa were detected, but

**Table 4** Impact of amoxicillin with clavulanic acid on the gastrointestinal microflora.

| Study | Dose (mg/day) | Days of administration | Number of patients | Impact on | | | | | | | | | Overgrowth | |
| | | | | Aerobic GP cocci | Pseudomonas | Enterobacteria incl. E. coli | Anaerobic cocci | Lactobacillus | Bifidobacteria | Bacteroides | | Clostridium | Candida |
|---|---|---|---|---|---|---|---|---|---|---|---|---|---|
| Lode et al (2001)[51] | 875/125 | 7 | 12 | Increase | – | Increase | Little change | Decrease | Decrease | Little change | | Decrease | |
| Young & Schmidt (2004)*[49] | 875/125×2 | 10 | 1 | – | – | Clusters present | – | – | Clusters present | Clusters present | | Two clusters | |
| Engelbrektson et al (2009)†[50] | 875/?x2 | 7 | 32 | – | – | Increase | – | No trend | No trend | Increase | | No trend | |
| Forssten et al (2014)†‡[48] | 875/125 | 7 | 80 | | | Increased | | Not affected in probiotic group | Decreased in the placebo group | | | Decreased | |
| Korpela et al (2016)‡[31] | – | – | 142 | | | | | Slight decrease | | Two fold elevated Parabacteroides | | | |
| Kabbani et al 2017[52] | 875/125×2 | 7 | 12 | Increase | | Increased | | Decreased | Decreased | Increased | | Decreased | |
| Mangin et al (2012)[47] | 875/125×2 | 5 | 18 | | | Increased | | | Decreased | | | | |

*This paper reported shifts in the representation of the major bacterial groups.
†Probiotic also given.
‡And penicillin V.
GP, gram positive.

2 weeks after cessation of antibiotics, there was a reappearance of this cluster. In this case study, the decrease in this cluster may be linked to the antibiotic-associated diarrhoea.

One arm of a study by Engelbrektson et al[50] observed the effects of probiotic treatment concurrent with antibiotic therapy on faecal communities. They analysed their data by measuring individual divergences from baseline levels after treatment to address the problem of subject-to-subject variability. Subjects fell into two categories: those with a stable baseline microbiota and those where it varied significantly. Culture data showed increasing numbers of *Bacteroides* and Enterobacteriaceae, but no trend for *Clostridium*, *Bifidobacterium* and *Lactobacillus*. The antibiotic group (without probiotic) had a significant change in numbers of bacteria from baseline at day 21.

Forssten et al[18] also studied antibiotic administration in combination with probiotic. Consumption of the probiotic combination mainly led to an increase in the faecal levels of the species included in the preparation. The antibiotic had only minor effects on *Lactobacillus*, *Bifidobacterium*, *Bacteroides*, Enterobacteriaceae and *Clostridium*.

Korpela et al[31] looked at the macrolides amoxicillin with and without clavulanic acid and penicillin V. Macrolide use reduced the abundance of Actinobacteria and increased Bacteroidetes and Proteobacteria. Penicillin groups did not have distinctly different phyla composition. Firmicutes was reduced. The effect of macrolide use was long lasting and was associated with increased risk of asthma and antibiotic-associated weight gain.

Twelve healthy subjects received amoxicillin/clavulanate for 7 days and their stool specimens were analysed using 16s rRNA gene pyrosequencing.[52] Antibiotic-associated changes included reduced prevalence of the genus *Roseburia* and increases in *Escherichia*, *Parabacteroides* and *Enterobacter*. Microbiota alterations reverted toward baseline, but were not completely restored 2 weeks after treatment.

Real-time PCR with temporal temperature gradient gel electrophoresis (TTGE) showed that at the end of a 5-day course of amoxicillin-clavulanic acid, total bacterial and Bifidobacteria concentrations were significantly reduced in 18 healthy volunteers.[47] At the same time, the mean similarity percentage profiles were significantly reduced and the number of distinct *Bifidobacterium* species per sample significantly decreased. Two months after antibiotic exposure, the mean similarity percentage had not normalised.

### Nitrofurantoin

Three studies investigated the effect of nitrofurantoin macrocrystals on faecal microbiota compositions.[53–55] Two studies used 16S rRNA gene sequencing and one study bacterial culture and these are summarised in table 2. Nitrofurantoin inhibits bacterial DNA, RNA and cell wall protein synthesis. It is used prophylactically as a urinary anti-infective agent against most gram-positive and gram-negative organisms and for long-term suppression of infections. On oral administration, most of the nitrofurantoin is rapidly absorbed in the small intestine and eliminated primarily by kidney glomerular filtration into urine, where it reaches higher and more effective therapeutic concentrations. Only a small amount reaches the colon, which may account for the minor impact on intestinal microflora.

One of the studies investigated a small population of women (n=7) with recurrent UTI and it showed that nitrofurantoin did not alter either the Enterobacteria, Enterococci or yeasts of the colonic flora during treatment.[55] No resistant strains of Gram-negative aerobic bacteria were detected.

Stewardson et al[53] treated patients with UTI (n=10) with nitrofurantoin macrocrystals and showed that it was not associated with a statistically significant global impact on the gut microbiota (weak effect). Nitrofurantoin treatment was associated with an increase in the proportion of the genus *Faecalibacterium* and a decrease in the proportion *Clostridium* (Clostridiaceae) at the end of the antibiotic treatment. Another small study (n=8) treated uncomplicated UTIs and did not show any significant impact of nitrofurantoin treatment on the faecal microbiota other than a temporary increase in the Actinobacteria phylum, more specifically in the beneficial *Bifidobacterium* genus.[54] Bacterial abundance had returned to pre-antibiotic levels 31 to 43 days after stopping antibiotic treatment, while the other study did not do follow-up testing.[53 54]

### Doxycycline

Four studies reported on the effect of doxycycline on faecal microbiota, one at suboptimal dosage (20 mg for 9 months) in patients with periodontitis[57] and two at usual dose 100–150 mg for 7–10 days, although one also with a probiotic,[56 58] and another at low dose (40 mg for 16 weeks),[59] as described in table 2. Doxycycline treatment did not significantly affect counts of total anaerobic bacteria, candida, total enterics, *Staphylococcus* or doxycycline-resistant bacteria recovered at any of the sample periods and did not result in the development of multi-antibiotic resistance.[57] Matto et al[58] specifically evaluated the influence of doxycycline therapy on the composition and antibiotic susceptibility of intestinal Bifidobacteria in nine subjects while they were also taking a probiotic and compared these to adults consuming only probiotics. A marked decrease in diversity of *Bifidobacterium* populations was observed during doxycycline therapy. Tetracycline-resistant *Bifidobacterium* isolates were more commonly detected in the antibiotic group than in the control group, thus increasing the pool of resistant commensal bacteria in the intestine.

In another study on 10 healthy volunteers, doxycycline decreased the abundance of Enterobacteriaceae, *Enterococcus* spp, *E. coli* and *Streptococcus* spp.[56] With the exception of *Fusobacterium* spp, which was eliminated, the number of anaerobic bacteria in faeces was not influenced by doxycycline.[56] After doxycycline administration, bacterial abundance was reported to have returned to pre-antibiotic levels 9 days after stopping

treatment.[56] The fourth study also looked at a low dose over a period of 16 weeks.[59] There were 2 log decreases in the numbers of enterococci and *E. coli*. Other aerobic microorganisms including enterobacteria, *Candida* spp, were not affected. There were no significant changes in the numbers of anaerobic lactobacilli, bifidobacteria, clostridia and *Bacteroides* during doxycycline administration.[59] No *C. difficile* strains were isolated. The aerobic and anaerobic microflora was normal at 28 days after stopping treatment.[59] In summary, doxycycline interferes with a microorganism's ability to manufacture proteins. At suboptimal dosage, it has little effect on the gut microbiota with the exception of enterococci and *E. coli*. At normal dosage it markedly affects the diversity of Bifidobacteria populations in one study and eliminated *Fusobacterium* sp in another. The effect on populations of other gut bacteria seems transient and normalisation was reported by 28 days. Longer term effects are not known.

### Clarithromycin

There were four articles on influence of clarithromycin included. Three were from the same research institute, and these examined the intestinal microflora of healthy volunteers before, during and after administration of clarithromycin. The study design was similar in all three studies and involved small numbers, 10 (a mix of male and female subjects) in two studies[60 62] and 12 men only[61] as described in table 2. Dosing and faecal sampling strategy and analysis were also similar, 250 mg twice daily for 7 days,[59] 500 mg twice daily for 7 days,[60] and 250 mg twice daily for 10 days,[62] sampling three times during and three to four times after administration up to 35 days. The male-only subjects were administered another antibiotic with a 6-week washout period; it is not clear whether this was before or after clarithromycin administration.

The main impact in the earlier study was a reduction in the numbers of Enterobacteria and Streptococci. Lactobacilli, Bifidobacteria and Bacteroides were suppressed in the anaerobic microflora.[60]

Clarithromycin caused a significant reduction of *E. coli* during 7 days of treatment but levels returned to normal 28 days after the end of the course.[61] Six subjects were colonised by resistant aerobic Gram-negative rods, *Citrobacter*, *Klebsiella*, *Proteus* and *Pseudomonas*. The total numbers of anaerobic bacteria decreased. Bifidobacteria and Bacteroides were significantly reduced, and Lactobacilli and Clostridia were suppressed but this was not significant. There was no overgrowth of *C. difficile* or yeasts. The microflora returned to normal in all subjects after 35 days.

In the final study, the numbers of *E. coli* were significantly reduced, and in six individuals, overgrowth of *Klebsiella*, *Citrobacter* and *Enterobacter* spp occurred.[62] The total number of anaerobic intestinal bacteria was not affected, but the numbers of Lactobacilli and Bifidobacteria were significantly reduced which persisted for the duration of the study. There was no significant overgrowth of *Candida* spp and no subjects were colonised by *C. difficile*.

A fourth study from a different group on six healthy volunteers showed no change in the abundance of *Candida* spp, *Enterococcus* spp or anaerobic bacteria. There was no new colonisation by *C. difficile*. Numbers of Enterobacteriaceae decreased slightly. Levels were reported to have returned to baseline by day 21 after the course of clarithromycin.

To summarise, all four studies reported reduction in Enterobacteria and three reported suppression of anaerobic bacteria after administration of clarithromycin. This reduction was transient for most species, except Lactobacilli and Bifidobacteria. Only one studied reported isolation of *C. albicans*.[60] None reported new colonisation or overgrowth of *C. difficile*. All reported normalisation to baseline levels within 28 days of finishing antibiotic treatment.

### Phenoxymethylpenicillin

Phenoxymethylpenicillin did not have much effect on gut microbiome, as shown by two studies that investigated the effect on the oropharyngeal and intestinal microflora of healthy volunteers (table 2).[64 65] A very early study reported no effect of phenoxymethylpenicillin on abundance of various species comprising the aerobic and anaerobic flora of faecal samples.[64] A later study showed no significant alterations in the total aerobic and anaerobic of the intestinal microflora were observed, although three volunteers became newly colonised with *Klebsiella* sp and one harboured high numbers of a non-fermentative Gram-negative rod. The numbers of viridans streptococci, enterococci and *Bacillus* were unaffected by the administration of phenoxymethylpenicillin, while minor alterations were noticed in the numbers of *E. coli*. The median values of *Clostridium* species increased during administration, while the numbers of *Bacteroides* species were unaffected during the study period. The microflora became normalised 2 weeks after withdrawal of the drugs.

### Erythromycin

In an early study,[66] it was shown that the administration of 500 mg of erythromycin twice daily for 7 days to 10 volunteers resulted in decreased abundance of both aerobic and anaerobic faecal flora. In addition, potentially pathogenic erythromycin-resistant enterobacteria, clostridia or yeasts colonised all subjects.[66] Brismar and colleagues[60] administered erythromycin orally for 7 days to 10 volunteers and evaluated its effect on the colonic bacteria. They found decreases in the numbers of streptococci, enterococci and enterobacteria during administration, increases in staphylococci and alteration of the anaerobic bacteria.

## DISCUSSION
### Main findings

This systematic review examined the use of the most commonly prescribed antibiotics in primary care to treat RTI and UTI and their impact on gut microbiota.

First, the studies showed that antibiotics have an impact on the abundances of the bacteria in the gut community, causing rapid and diminished levels of bacterial diversity and taxonomic richness, increases and decreases in the relative abundances of certain taxa, leading to dysbiosis, as well as antibiotic and individual host-specific effects.

Second, once treatment has stopped, there was some, but poor, evidence that the gut bacteria show resilience and are capable of some degree of recovery, in most individuals, to their initial state. However, the microbiota is often not totally recovered, suggesting some antibiotics have a persistent effect on certain species. These observations underline the importance of restrictive and proper use of antibiotics in order to prevent long-term ecological disturbances of the indigenous microbiota.

### Comparative observations

Other reviews have also reported and summarised how antibiotics change the abundance and diversity of the intestinal microbiota.[68–70] Here, we focus on those antibiotics routinely prescribed by general practitioners (GPs) in primary care. Often GPs inappropriately prescribe antibiotics for infections caused by viruses, or prescribe a broad-spectrum antibiotic, when an antibiotic for a specific bacteria should be used. They may also prescribe the incorrect dose or for the wrong length of time. Health professionals are concerned that antibiotics are used too often and incorrectly, which contributes to antimicrobial resistance and with the additional disruption to the microbiome also can have other negative (long term) health effects, such as metabolic and immune disorders.[71 72] These additional effects will further impact on the patient and potentially on the primary care burden. Inappropriate prescribing is due to many factors including patients who insist on antibiotics, GPs who do not have enough time to explain why antibiotics are not necessary those who do not know how to recognise a serious bacterial infection or those who are overly cautious.

Quantifying antibiotic therapy effect on gut flora is challenging. Although the literature describes 'normal', it really means those commensal species which are consistently and predominantly found in the healthy human gut. Studies quantifying changes in flora would be easier to interpret if a baseline populations were established in each case. We found a lack of large-scale trials (including RCTs) and observational studies and heterogeneity regarding methodology and outcomes. The heterogeneity is apparent by the inconsistency in the dosage, duration of treatment and follow-up, selection and spectrum of antibiotics used, all contributing to varying faecal concentrations. The health status and age of the participant also have an effect, most studies did not clearly define baseline characteristics, which causes problems with comparing outcomes. The majority of studies did not report adverse events or any other factors which might affect the efficacy of the antibiotic which

in turn might influence the outcome on the microbiome. The studies included used different methods, for example, random allocation of antibiotics to healthy volunteers, or non-randomised treatment where all healthy subjects in the study received antibiotics, also some studies participants were masked to which treatment they received and others were not. There were differences between studies in the way outcomes are defined and measured, for example, culturing versus molecular techniques. These differing analytical techniques may lead to differences in the observed intervention effects. A number of the included studies originate from one research group which could add additional bias to the outcomes. The different classes of antibiotics also have different effects on the microbiome, making it difficult to compare across studies.

The composition of the gut microbiota among humans varies considerably from subject to subject. Grouping of data from several individuals can result in loss of statistical significance. The studies do not consider any confounding factors such as diet, age of subjects, childhood exposure to antibiotics, geographical location, stress or taking probiotic supplements, all of which can complicate the specific antibiotic treatment effects on the gut microbiome profile. These factors can change over a lifetime.

### Strengths and limitations

The primary strength of this study is our focus on the most commonly prescribed antibiotics in primary care and their effect on gut microbiota. Information on how these commonly prescribed antibiotics affect our gut microbiome could influence GP decision-making when prescribing antibiotics. Also, the adverse impact of antibiotics on the gut microbiome is of great importance because when disrupted, it can be associated with a variety of diseases, including susceptibility to infections, autoimmune diseases (such as inflammatory bowel disease), diabetes, depression and obesity. The main limitations of this review are that the majority of the studies included were small, not properly randomised nor they were large observational studies. The considerable heterogeneity is prone to biases and confounding factors.

Inherent in many studies that address the impact of antibiotics on the intestinal microbiota are the limitations of techniques available for analysis. Many have been performed using laborious culture-based methodology which do not lend themselves to analysis of large numbers of samples. The older studies included in our analysis all used culture-based techniques to examine gut microbiota composition, which increases the risk of detection bias. The disadvantage of using culturing is that despite the use of specific selective media and anaerobic incubation conditions, there remains a substantial part of the microbiota, approximately 80%,[67] that has not yet been cultured. Culture media choice and sample handling can also skew the data.

The limitations of culture-based techniques can be largely overcome by using molecular approaches.[73] These methods are based on 16S rRNA gene amplification and give us a broader less biassed view of gut bacterial composition. The amplified genes are characterised by methods, such as terminal-restriction fragment length polymorphism, denaturing or temperature gradient gel electrophoresis, deep shotgun metagenome and full-length sequencing, some of which are used in the more recent papers in our review. These bioinformatic techniques also have limitations in that they are often more difficult to understand and interpret. The use of different techniques makes it more difficult to compare results, and generalisation of conclusions is a result of this diversity in determination and quantification.

## CONCLUSIONS

The widespread and often overuse of antibiotics has led to the establishment of antibiotic-resistant bacteria and the transfer of resistance genes, which has become a global challenge for infection control and the untreatable nature of bacterial infections. Almost 80% of the antibiotics prescribed within the National Health Service are within primary care.[1] The effects of excessive antibiotic exposure can also be seen in the symbiotic microbiotas of the human body.[74] As a result, the microbiota imbalances caused by antibiotics can negatively affect health by increasing susceptibility to infections, compromising immune homeostasis (indicative in increased allergies), asthma, seen by obesity, metabolic syndrome and diabetes.[75]

Antibiotics have an impact on the gut microbiota, causing rapid and diminished levels of bacterial diversity and increases and decreases in the relative abundances, leading to dysbiosis. Once treatment has stopped, there was some evidence that the gut bacteria are capable of some degree of recovery, in most individuals, to their baseline state. The studies do not consider that recovery of the microbiome after antibiotic therapy in the elderly may be affected by age-associated physiological alterations and other drug–drug interactions.

This systematic review highlights the lack of research and understanding of the effects of commonly prescribed antibiotics on the gut microbiome and shows it is an area that requires studies on larger populations with extended sampling to understand the long-term impacts. Embedding of this knowledge in antimicrobial stewardship programmes in primary care will be essential.

**Contributors** KTE and VW performed the searches. VW, ETvdW, AH, LD and KTE identified eligible studies. KTE, VW and ETvdW appraised study quality and extracted data. KTE and VW drafted the first sections of the text and KTE completed data analysis, text and tables to circulate to all authors for their contribution to the final draft. ALH advised on search strategy and RoB and read and contributed to the draft. ADH was involved in the main concept, advised on the antibiotics included and read and contributed to the draft. ETvdW is the guarantor and attests that all listed authors meet authorship criteria and that no others meeting the criteria have been omitted. All authors had full access to the data in the study and can take responsibility for the integrity of the data and the accuracy of the data analysis.

**Funding** This research was supported by the NIHR Health Protection Research Unit in Evaluation of Interventions at the University of Bristol. The views expressed in this article are those of the author(s) and not necessarily those of the NHS, the NIHR or the Department of Health and Social Care.

**Competing interests** All authors had financial support from the University of Bristol for the submitted work; no financial relationships with any organisations that might have an interest in the submitted work in the previous three years; no other relationships or activities that could appear to have influenced the submitted work.

**Patient consent for publication** Not required.

**Provenance and peer review** Not commissioned; externally peer reviewed.

**Data availability statement** Data are available upon reasonable request. All data relevant to the study are included in the article or uploaded as supplementary information. Data sharing. If necessary data not included in the paper can be shared.

**ORCID iDs**
Karen T Elvers http://orcid.org/0000-0001-5345-8662
Ashley Hammond http://orcid.org/0000-0002-6657-1514
Alyson L Huntley http://orcid.org/0000-0001-9409-7891
Esther T van der Werf http://orcid.org/0000-0002-5330-5735

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
