## [Reviewer comments · BMJ Open]

ARTICLE DETAILS

TITLE (PROVISIONAL)	Antibiotic-induced changes in the human gut microbiota for the most commonly prescribed antibiotics in primary care in the United Kingdom: systematic review
AUTHORS	Elvers, Karen; Wilson, Victoria; Hammond, Ashley; Duncan, Lorna; Huntley, Alyson; Hay, Alastair; van der Werf, Esther

VERSION 1 – REVIEW

REVIEWER	Quentin LE BASTARD University of Nantes, France
REVIEW RETURNED	21-Jan-2020

GENERAL COMMENTS	The impact of medications on the intestinal microbiota is a complex issue. Of all those that we can prescribe, antibiotics are probably the ones that have the strongest action on the digestive flora, raising concerns about the risk of emergence of resistant bacteria. Authors choose to focus on antibiotics commonly prescribed in primary care. This approach seems really relevant. Here, authors provide a list of actions on several genii and taxa for the most commonly used molecules. As they pointed out, our knowledge in this area is very limited. It is difficult to draw conclusions from the current literature given the heterogeneity of protocols and methods of analysis of the microbiota. This is currently a strong limitation that prevents us from making recommendations for clinical practice. The authors pointed out all these limitations.
---

REVIEWER	Annie S. Hong University of Nevada Las Vegas School of Medicine
REVIEW RETURNED	28-Mar-2020

GENERAL COMMENTS	Systematic review is useful data but requires major revisions. 1. How does this study differ from study "The effect of antibiotics on the composition of the intestinal microbiota - a systematic review" published in December 2019 (DOI: https://doi.org/10.1016/j.jinf.2019.10.008). Would be recommended to reference and address this similar study. In what way is your study different? What further data does it provide? What accounts for the difference in outcomes and number of qualifying studies between the two studies? 2. Purpose/Methods/Results are clear and well supported. Limitations is comprehensive but should justify more why all studies regardless of risk of bias. For example, one study only has
--

	n=1. Iff the data is limited and this is clearly stated that is still acceptable. 3. Some abbreviations are not consistent or used correctly. For example: -Line 476 "We found a lack of large-scale trials (RCTs)" - Line 484 "... could influence GPs decision making in antibiotic prescribing in primary care." 4. Please have numerous authors do edits. There are many mistakes, a few examples: - Line 375 "A macrolides amoxicillin with and without" - Line 398 "Nitrofurantoin treatment was associated with an increase in the proportion of the genus Faecalibacterium;and decrease in the proportion Clostridium (Clostridiaceae) at the end of the antibiotic treatment" - Line 418 "At normal dosage it markedly effects.." - Line 465 "..leading to dysbiosis. As well as antibiotic- and individual host-specific effects."
--	---

VERSION 1 – AUTHOR RESPONSE

Reviewer(s)' Comments to Author:

Reviewer: 1

Reviewer Name:

Quentin LE BASTARD

Institution and Country:

University of Nantes, France

Please state any competing interests or state 'None declared':

None declared

Please leave your comments for the authors below:

The impact of medications on the intestinal microbiota is a complex issue. Of all those that we can prescribe, antibiotics are probably the ones that have the strongest action on the digestive flora, raising concerns about the risk of emergence of resistant bacteria. Authors choose to focus on antibiotics commonly prescribed in primary care. This approach seems really relevant. Here, authors provide a list of actions on several genii and taxa for the most commonly used molecules.

As they pointed out, our knowledge in this area is very limited. It is difficult to draw conclusions from the current literature given the heterogeneity of protocols and methods of analysis of the microbiota. This is currently a strong limitation that prevents us from making recommendations for clinical practice.

The authors pointed out all these limitations.

Reviewer: 2

Reviewer Name:

Annie S. Hong

Institution and Country:

University of Nevada Las Vegas School of Medicine

Please state any competing interests or state 'None declared':

None declared

Please leave your comments for the authors below:

Systematic review is useful data but requires major revisions

1. How does this study differ from study "The effect of antibiotics on the composition of the intestinal microbiota - a systematic review" published in December 2019 (DOI:

<https://doi.org/10.1016/j.jinf.2019.10.008>). Would be recommended to reference and address this similar study. In what way is your study different? What further data does it provide? What accounts for the difference in outcomes and number of qualifying studies between the two studies?

We have referenced this study (which was not published when we first submitted this review) and revised our text to clarify our rationale and findings. Our study is different because it focuses on specific antibiotics prescribed in primary care and addresses GPs concerns regarding contribution to antimicrobial resistance and inappropriate prescribing, there is increasing evidence that these antibiotics are often prescribed by GPs where the cause of infection, viral or bacterial, is unknown or for conditions which are mild enough not to be treated. Both of these scenarios lead to over prescribing and the associated problem of antimicrobial resistance. Any resulting disruption of the microbiome is likely to cause additional (long term) health problems that are potentially more significant than the initial infection. This review provides additional articles on these specific antibiotics that were not included in the 2019 Systematic Review. Differences in perceptions about how risk-free antibiotic treatment is, may in part, account for the enormous variation in rates of their use from practitioner to practitioner but also between a specific group of practitioners such as GPs. The real costs of the most prescribed antibiotics in primary care must be clearly understood, including the differences between particular antimicrobial agents in their effects on the microbiome. This review provides information specifically useful for antimicrobial stewardship in primary care. The difference in qualifying studies is two-fold, a different search strategy was used including three databases. Different inclusion criteria were used, primarily restricting the search to the specific antibiotics which are most commonly prescribed in general practice.

2. Purpose/Methods/Results are clear and well supported. Limitations is comprehensive but should justify more why all studies regardless of risk of bias. For example, one study only has n=1. If the data is limited and this is clearly stated that is still acceptable.

Updated RoB to include additional studies. We included all studies in our risk of bias assessment as they all fit our inclusion criteria. We reported this risk of bias in the narrative discussion of the heterogeneity in the studies, and felt it important to include all studies to highlight the problems with the comparability of diverse studies.

3. Some abbreviations are not consistent or used correctly. For example:

-Line 476 "We found a lack of large-scale trials (RCTs)"

Adjusted this to say that this includes RCTs.

- Line 484 "... could influence GPs decision making in antibiotic prescribing in primary care."

We have defined our use of GP and GPs.

4. Please have numerous authors do edits. There are many mistakes, a few examples:

- Line 375 "A macrolides amoxicillin with and without"

looked at macrolides amoxicillin

- Line 398 "Nitrofurantoin treatment was associated with an increase in the proportion of the genus Faecalibacterium;and decrease in the proportion Clostridium (Clostridiaceae) at the end of the antibiotic treatment"

Nitrofurantoin treatment was associated with an increase in the proportion of the genus

Faecalibacterium ;and a decrease in the proportion Clostridium (Clostridiaceae) at the end of the antibiotic treatment.

- Line 418 "At normal dosage it markedly effects.."

At normal dosage it markedly affects

- Line 465 "..leading to dysbiosis. As well as antibiotic- and individual host-specific effects."

As well as antibiotic and individual host-specific effects